# Research on the Preparation and Application of Synthetic Leather from Coffee Grounds for Sustainable Development

**Yujing Tian** [1] 🔲 **, Jinyu Wang** [1] **, Sixian Zheng** [2] **, Xinyue He** [1] **and Xiaogang Liu** [1,*]

1   College of Fashion and Design, Donghua University, Shanghai 200051, China
2   Zhejiang ZAKER New Material Technology Co., LTD., Lishui 323010, China
*   Correspondence: liuxg@dhu.edu.cn

**Abstract:** As the market demand for environmentally friendly synthetic leather products has increased, water-based synthetic leather manufacturing technology and product performance have made great progress. Along with the explosive growth of coffee grounds generated by urban consumers in their daily lives, research on the sustainable reuse of coffee grounds has gradually become a trend in the field. This study discusses the method of preparing environmentally friendly water-based synthetic leather that reuses coffee grounds and is assessed by standardized physical tests for friction color fastness, Martindale abrasion resistance, breathability and moisture permeability, softness, and peel strength. The results have indicated that sustainable coffee-ground synthetic leather fully meets the performance of aqueous synthetic leather for apparel and luggage, with even some performance indicators exceeding existing aqueous synthetic leather, which is an innovative and sustainable product that can be applied to the apparel industry in the future. Its development and application in the textile field will provide research ideas with the transformation of environmental problems into new opportunities.

**Keywords:** sustainable; coffee grounds; synthetic leather; material preparation; application research





## 1. Introduction

As a composite material that can replace animal leather, synthetic leather has better durability and high-cost performance compared to animal leather, which makes it highly acceptable in the consumer market [1]. Synthetic leather is widely used to make shoes, boots, garments, bags, [2] etc. However, the traditional solvent-based synthetic leather manufacturing process cannot avoid organic solvent pollution, and its finished products have potential chemical hazards to users [3,4]. Along with the increase in market demand for environment-friendly synthetic leather products, water-based synthetic leather manufacturing technology and product performance have made great progress. It is widely concerned by society and enterprises, and the development of green and clean production of synthetic leather has become an important research direction in the process of continuous research and application, and it has been gaining market share in recent years.

At the same time, with the continuous improvement of the living standards and consumption ability of consumers, the waste generated in daily life, such as coffee grounds, bagasse, glass residue, discarded shells, and other waste, has also seen explosive growth, bringing a great burden to urban cleaning. According to data released by the International Coffee Organization, more than 2 billion cups of coffee are sold daily worldwide, generating an average of 40,000 tons of coffee residue daily. If coffee grounds are landfilled as conventional waste, they will produce a large amount of methane, one of the "greenhouse gases", which will pollute soil and water resources [5]. However, the waste coffee grounds still retain a high-reuse value and are explored in the fields of textile yarns, printing and dyeing processes, and automotive parts [6,7]. In addition, coffee grounds have been utilized to synthesize advanced materials for environmental sustainability, such as photocatalysts and photothermal materials [8,9].

At present, the synthetic leather industry is facing a situation where the level of clean production needs to be improved and the market's competition pressure is increasing. The overall market and research field are still short of sustainable experimental research on the resourcefulness of tannery solid waste and clean production processes [10], and the product development is also of a relatively basic design and application. In view of this, this paper focuses on the shortcomings of the existing technology to explore the preparation of sustainable coffee grounds synthetic leather and to develop an environmentally friendly water-based synthetic leather that can effectively reuse the waste coffee grounds in daily life without potential chemical hazards to users. In addition, the feasibility of the process was verified by testing the rubbing color fastness, the Martindale rub test, vapor permeability, and softness of the samples under different conditions. By analyzing the properties of this material over traditional synthetic leather, and discussing its application and design methods for clothing and accessories, and using appropriate surface finishing and combination processes, which can meet the personalized demand of consumers for sustainable materials and also innovate the industry for green and sustainable production and processing of synthetic leather. The supplier of the material related to this paper is Zhejiang ZAKER New Material Technology Co., Ltd., Lishui, China.

## 2. Material and Methods

### 2.1. Raw Materials and Composition

The composition of eco-friendly coffee-ground synthetic leather includes top layer, middle layer, and base layer in order, from outside to inside [11].

2.1.1. Raw Material Composition of the Surface Layer

Waterborne polyurethane top-coat slurry coating, slurry configuration, including waterborne polyurethane emulsion, defoamer, solid waste slag, fungicide, antimold agent, crosslinking agent, wetting agent, emulsifier, thickening agent. The specific component content is shown in Table 1. The mass fraction in the content of ingredients is the percentage of the total mass of a substance in a mixture.

**Table 1.** Water-based slag topping slurry configuration.

| No. | Ingredients | Content of Ingredients |
| --- | --- | --- |
| 1 | Waterborne polyurethane emulsion | 40–60% |
| 2 | Defoamer | 0.2–1% |
| 3 | Solid waste residue | 10–80% |
| 4 | Fungicide | 0.5–2% |
| 5 | Antimold agent | 0.5–2% |
| 6 | crosslinking agent | 0.2–3% |
| 7 | Wetting agent | 1–2% |
| 8 | Emulsifier | 0.1–3% |
| 9 | Thickening agent | 2–3% |

2.1.2. Raw Material Composition of the Middle Layer

The raw material of the intermediate layer can be divided into three different composition forms: (1) one adhesive layer, (2) two foaming layers, or (3) one foaming layer with one adhesive layer. Wherein, the said foaming layer is formed by coating it with an aqueous residue foaming layer. The foaming layer slurry configuration includes waterborne polyurethane emulsion, foaming agent, solid waste residue, fungicide, antimold agent, crosslinking agent, emulsifier, and thickener. The adhesive layer is obtained by coating it with an aqueous residue adhesive layer slurry, wherein the said aqueous residue adhesive layer slurry configuration includes waterborne polyurethane emulsion, solid waste residue, fungicides, antimold agents, crosslinking agents, wetting agents, emulsifiers, and thickeners. The aqueous residue is created by grinding coffee grounds through low

temperature liquid nitrogen and then adding waterborne polyurethane resin, which is then dispersed by high-speed shear. The specific component content is shown in Table 2.

**Table 2.** Water-based slag foam layer and water-based slag bonding layer slurry configuration.

| Water-Based Slag Foam Layer Slurry | | Water-Based Slag Bonding Layer Slurry | |
| --- | --- | --- | --- |
| Ingredients | Component Content | Ingredients | Component Content |
| Waterborne polyurethane emulsion | 40–60% | Waterborne polyurethane emulsion | 40–60% |
| Foaming agent | 3–6% | Solid waste residue | 10–80% |
| Solid waste residue | 10–80% | Fungicide | 0.5–2% |
| Fungicide | 0.5–2% | Antimold agent | 0.5–2% |
| Antimold agent | 0.5–2% | Crosslinker | 0.2–3% |
| crosslinking agent | 0.2–3% | Wetting agent | 1–2% |
| Emulsifier | 0.1–3% | Emulsifier | 0.1–3% |
| Thickening agent | 2–3% | Thickening agent | 2–3% |

### 2.1.3. Raw Material Composition of the Base

The base layer is a base cloth or water-based slag material base. The base fabric is impregnated in an impregnation tank with an impregnating solution (one or a combination of nonfluorinated water repellent, salt solution, and acid solution) and then extruded, ironed, and dried. Water-based slag base is either a water-based slag dry base or a water-based slag wet base. A water-based crumb dry base is created by impregnating the base fabric with an impregnating tank containing an impregnating solution (one or more combinations of fluorine-free water repellent, salt solution, and acid solution), coating with water-based crumb foam layer slurry, drying, and winding. Water-based slag wet base consists of base cloth impregnated with an impregnating solution (one or more of surfactant, nonfluorinated water repellent, salt solution, acid solution). After impregnation in the impregnation tank, it is coated with a water-based slag foam layer slurry, solidified in a solidification tank containing a solidification solution, then dried and washed, and then dried and wound.

The particle size of the solid waste residue material described above is 0.1–5 μm. Preferred specific components of specific substances are supplemented as shown in Table 3.

**Table 3.** Supplementary Table of Preferred Specific Components.

| Ingredients | Specific Substances |
| --- | --- |
| Fungicide | Isothiazolinone fungicide. |
| Waterborne polyurethane emulsion | One or more combinations of anionic aromatic waterborne. Polyurethane emulsions, anionic aliphatic waterborne polyurethane emulsions, cationic aromatic anionic, or cationic aliphatic waterborne polyurethane emulsions. |
| Defoamer | Silicone defoamer or mineral oil defoamer. |
| Foaming agent | One or more of organosilicon modified foaming agent, sodium dodecyl benzene sulfonate foaming agent, and stearate type foaming agent compounded. |
| Fungicide | Isothiazolinone fungicide; antimildew agent is o-phenylphenol sodium antimildew agent |
| Crosslinking agent | One or more of aziridine crosslinking agent, polycarbodiimide crosslinking agent, closed isocyanate crosslinking agent |
| Wetting agent | Silicone wetting agent or acetylene glycol wetting agent |
| Emulsifier | Nonionic emulsifier or anionic emulsifier |
| Thickening agent | One or more compound of polyurethane bonding thickener, acrylate alkali-soluble thickener, cellulose thickener |

*2.2. Production Processes*

The preparation process of sustainable coffee grounds synthetic leather as shown in Figure 1 can be summarized as the following steps: (1) Preparation of top layer slurry; (2) preparation of intermediate layer slurry; (3) preparation of substrate; (4) coating and assembly.

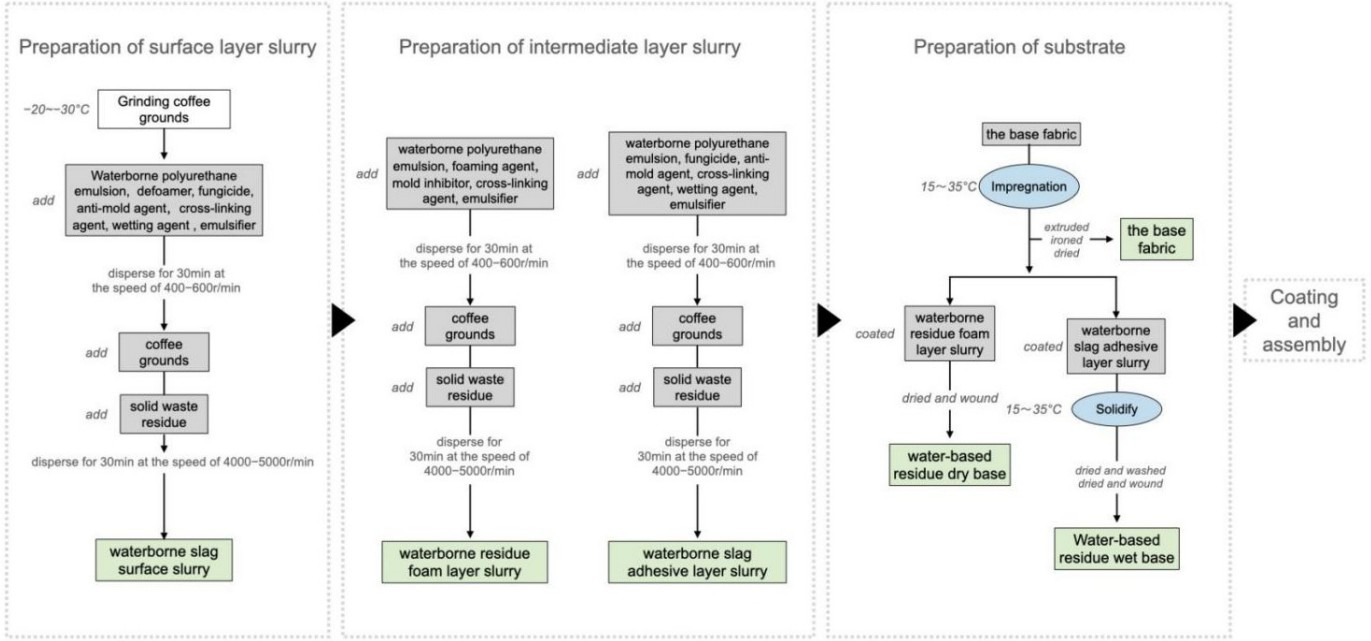

**Figure 1.** Sustainable coffee-ground synthetic leather preparation process.

2.2.1. Preparation of Surface Layer Slurry

Preparation of the topping slurry consists of the following two main steps.

Step 1: Grinding coffee grounds through a grinder at −20–30 °C degrees Celsius to a particle size of 0.1–5.0 μm.

Step 2: Preparation of waterborne slag surface slurry. Mix waterborne polyurethane emulsion (40–60%) and defoamer (0.2–1 part)according to the mass fraction, and add fungicide (0.5–2%), antimold agent (0.5–2%), crosslinking agent (0.2–3%), wetting agent (1–2%), emulsifier (0.1–3%), and disperse for 30 min at the speed of 400–600 rpm, then add solid waste residue (10–80%), and add thickener to adjust the viscosity of the slurry to 1000–5000 cps and disperse at 4000–5000 rpm for 30 min to get water-based slag surface slurry.

2.2.2. Preparation of Intermediate Layer Slurry

Preparation of the intermediate slurry consists of the following two main steps:

Step 1: Preparation of waterborne residue foam layer slurry. Add waterborne polyurethane emulsion (40–60%), foaming agent (3–6%), fungicide (0.5–2%), mold inhibitor (0.5–2%), crosslinking agent (0.2–3), and emulsifier (0.1–3%) and disperse at 400–600 rpm for 30 min and then add solid waste residue (10–80%). Add thickener to adjust the viscosity of the slurry to 5000–20,000 cps and disperse at 4000–5000 rpm for 30 min to obtain an aqueous slag foam layer slurry.

Step 2: Preparation of waterborne slag adhesive layer slurry. Add 40–60 parts of waterborne polyurethane emulsion, 0.5–2 parts of fungicide, 0.5–2 parts of antimold agent, 0.2–3 parts of crosslinking agent, 1–2 parts of wetting agent, and 0.1–3 parts of emulsifier, and disperse at 400–600 rpm for 30 min, and then add 10–80 parts of solid waste residue. Add thickener to adjust the viscosity of the slurry to 10,000–50,000 cps and disperse at 4000–5000 rpm for 30 min to get a water-based slag adhesive layer slurry.

2.2.3. Preparation of Substrate

The preparation of substrate consists of the following three main steps:

Step 1: Impregnation of the base fabric by a dipping tank containing an impregnating solution followed by extrusion, ironing, and drying.

Step 2: After impregnating the base fabric with the impregnating tank containing the impregnating solution, the base fabric is coated with a slurry of water-based residue foam layer, dried, and wound up to form a water-based residue dry base.

Step 3: After impregnation of the base fabric in an impregnation tank containing impregnating solution, it is coated with a water-based slagging foam layer slurry. It is solidified in a solidifying bath containing a solidifying solution, then dried in an oven and washed. It is then dried and wound to form the aqueous slag wet base.

The temperature of the impregnating solution is 15~35 °C and the temperature of the solidifying solution is 15~35 °C.

2.2.4. Coating Combinations

Depending on the preparation of the different surface layer slurry, intermediate layer slurry, and substrate described above, three different types of coating combination forms will be produced, as shown below.

(1)   The water-based slag surface slurry obtained in Section 2.2.1 is applied to the release paper and dried. Subsequently, the aqueous slag bonding layer obtained in Section 2.2.2 is coated and laminated to the base fabric or the base fabric obtained in Section 2.2.3, or the water-based residue dry base or the water-based residue wet base, and it is dried to obtain environmentally friendly water-based synthetic leather.

(2)   The aqueous slag top layer slurry obtained in Section 2.2.1 is coated on the release paper and dried. Subsequently, the aqueous slag foaming layer slurry obtained in Section 2.2.3 is passed through the foaming machine. Next is to adjust the foaming density to 0.1–1.0 kg/L, apply it to the top layer and dry it. Subsequently, repeat the coating of the aqueous slag foaming layer slurry, then laminate the base fabric or the base fabric obtained in step 5, or the aqueous slag dry base obtained in step 6 or the aqueous slag wet base obtained in Section 2.2.3 and dry to obtain environmentally friendly aqueous synthetic leather.

(3)   The aqueous slag top layer stock obtained in Section 2.2.1 is coated on the release paper and dried. The aqueous slag foaming layer obtained in Section 2.2.2 is then fed into the foaming machine. The foaming density is adjusted to 0.1–1.0 kg/L, applied to the top layer and dried. Subsequently, apply the waterborne slag adhesive layer, then laminate the base fabric or the base fabric obtained in 2.2.3 or the waterborne slag dry base or obtain the waterborne slag wet base and dry to obtain the environmentally friendly waterborne synthetic leather.

Since coffee grounds contain certain oils and grease, the ground residue is easily re-agglomerated with placement when added to waterborne polyurethane slurry, resulting in a gradual decrease in flatness during the slurry coating process in continuous production. Through the introduction of emulsifier, the preparation process, firstly, adjusts the dispersion of the slurry system for the oil-leaning residue components, ensuring that the residue powder can maintain a good dispersion for a short time. Secondly, the dispersibility of polyurethane particles in waterborne polyurethane resin is improved. This enables the resin to be stirred in a high-speed shearing environment, thus, solving the problem that the slag working slurry cannot be dispersed at high speed and ensuring the stability of continuous production.

*2.3. Process Advantages and Innovation Points*

1.   By using coffee grounds as a filler in the manufacturing process of water-based synthetic leather, the eco-friendly coffee grounds synthetic leather can effectively reuse the waste materials of daily life. The whole process does not require water and

does not use any acidic substances or animal-related ingredients, which fully satisfies sustainable properties.

2. The coffee grounds recovered after extraction have a special pore structure with a high-specific surface area. After effective treatment, it is added as a filler and combined with waterborne polyurethane using an aqueous production process to provide a synthetic leather product that is closer to the breathability of leather.

3. Waterborne polyurethane emulsion and related water-based additives are used as the main raw materials for synthetic leather manufacturing, and the corresponding products do not contain DMF. The product can completely solve the problem of chemical pollution caused by DMF in solvent-based synthetic leather in the market, and there is no potential chemical hazard to users. At the same time, the DMF recycling process is omitted, and the whole production process is energy-saving and environmentally friendly.

4. The natural coffee aroma of coffee and the fillings of coffee grounds produce a certain fragrance on the synthetic leather, giving an added value to the sensory of the synthetic leather product.

## 3. Analysis of the Application of Sustainable Coffee-Ground Synthetic Leather in Apparel Products

Due to the lack of natural leather resources, natural leather as a raw material for apparel products is more expensive. Synthetic leather, which is close to the structure of natural leather, can be developed artificially as an alternative. With the establishment of "green barriers" in international trade and the increasingly stringent international environmental requirements, the sustainability of the apparel manufacturing industry has become a conservative concern [12]. Therefore, in the face of the industry's demands for sustainable materials, traditional solvent-based synthetic leather is no longer able to effectively meet demands. Sustainable coffee-ground synthetic leather cannot only make sustainable use of coffee grounds, which is a waste of daily life, but also avoid the organic solvent pollution and potential chemical hazards to users that cannot be avoided in the manufacturing process of traditional solvent-based synthetic leather [13]. At the same time, the special pore structure of coffee grounds makes synthetic leather more breathable than genuine leather. Its innovative application in apparel products will improve the performance of traditional solvent-based synthetic leather products and can assist design methods to optimize the appearance of traditional products and enhance the added value. The coffee-ground synthetic leather is shown in Figure 2.

### 3.1. Analysis of the Types of Products Used

### 3.1.1. Use in Clothing

In the apparel field, artificial leather, which is distinct from natural leather, has become a widely used material in apparel design, but there has been a gap in the demand for sustainable materials [14,15]. Design often requires materials to be presented concretely, and the emergence of synthetic innovative materials with high performance and environmental attributes of coffee grounds can bring innovative potential to apparel design while meeting market demands. Designers plan all aspects of the material's color, structure, and surface texture through design orientation in the process of application to enhance synthetic leather garment design in a holistic manner. Therefore, the understanding and control of material characteristics are essential before material application.

Sustainable coffee-ground synthetic leather is used as a base material for garments, which can be laid out on collars and sleeves, as shown in Figure 3. By applying the material to the collar, which is close to the body's neck, the potential chemical hazards to the body can be avoided compared to traditional solvent-based products. Its high performance in vapor and moisture permeability can optimize comfort during wear. When applied to the sleeves, the garment is not stuffy or sticky when sweating. At the same time, the

slight coffee fragrance in the material can enhance the sensory dimension of the material experience and bring a storytelling connotation to the clothing products.

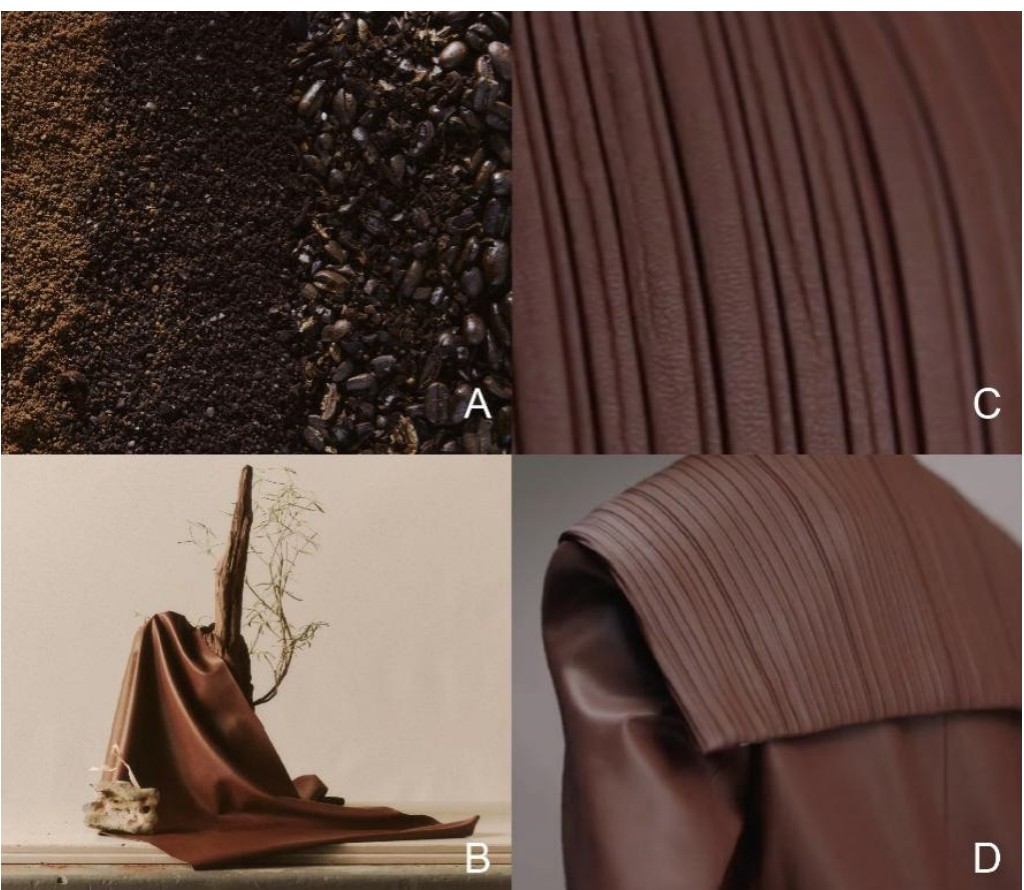

**Figure 2.** (**A**) Coffee grounds after recycling and grinding. (**B**) Sustainable coffee-ground synthetic leather products. (**C**) The surface process of coffee-ground synthetic leather: pleating. (**D**) A garment created with sustainable coffee-ground synthetic leather.

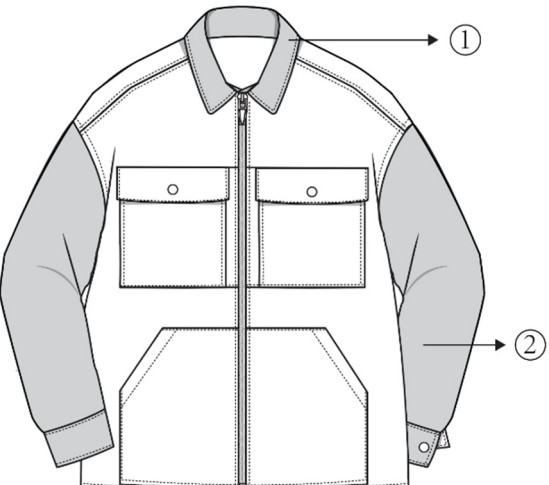

**Figure 3.** The use of coffee-ground synthetic leather in clothing. ① is a more suitable application for the collar area of the garment and ② is a more suitable application for the sleeve garment area.

### 3.1.2. Use in Clothing Accessories

Traditional luggage and footwear use natural leather as the raw material, which costs more. Under the development trend of man-made materials preparation and development, synthetic leather materials with high production efficiency and low cost are widely used [16]. However, traditional synthetic leather luggage and footwear products have disadvantages, such as poor breathability and are easy to crack and lose color after long-time use. The innovative sustainable coffee-residue synthetic leather material, when applied to the body of bags and the upper part of shoes, will enhance the sense of quality for apparel products with its better wear resistance compared to traditional synthetic leather and breathability close to that of natural leather, Suitable applications are shown in Figure 4.

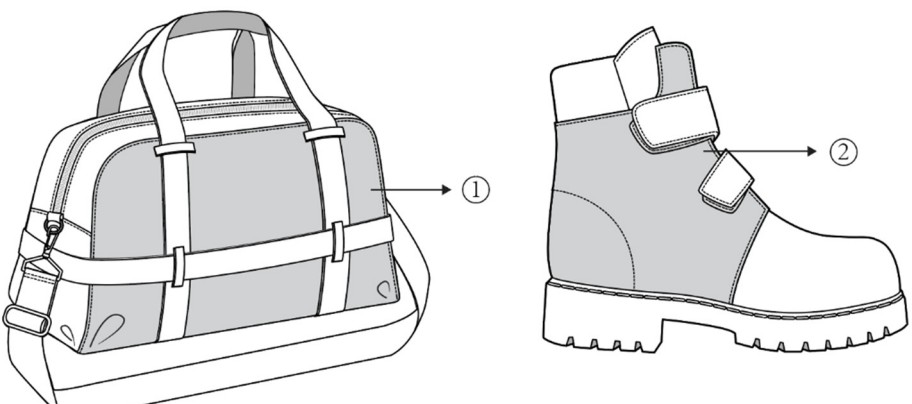

**Figure 4.** The use of coffee-ground synthetic leather in apparel products. ① is the main surface area of the bag and ② is the upper part of the shoe.

### 3.2. Specific Process Analysis

### 3.2.1. Surface Process

For the product composition of the same type of material, the designer can combine the material characteristics of coffee-ground synthetic leather and the design language in line with the product and adopt appropriate processing technology for the surface of the material. For example, hollowing, pleating, embossing, shaping, etc., [17] can produce diverse texture effects, change the original form of coffee-ground synthetic leather itself, and increase the sense of three-dimensionality and space. To make the material more abundant and harmoniously applied to the product, one can add details to the design performance of a single material, highlight the value of the product, and further extend the beauty of the applied product. For example, the sustainable property of coffee-ground synthetic leather material and its unique coffee fragrance can be used to amplify the surface technology design of the product and increase its added value on the basis of enriching the sensory experience of the product [18,19].

### 3.2.2. Combined Process

For application products composed of different materials, it is necessary to think about the overall visual performance of the product in the design stage and ensure that the addition of coffee-ground synthetic leather meets the coordination of material visuals in the application process. On this basis, the design can be carried out by means of splicing and weaving. For example, different textured materials can be cut and woven into grids to form a product with a soft color texture and change. In the process of splicing, it is necessary to consider the softness and thickness of the splicing materials to coordinate with the coffee-residue synthetic leather materials for selection, and at the same time, the color ratio and design also need to ensure the overall harmony of the product [20].

In summary, the application of sustainable coffee-ground synthetic leather in clothing products can take advantage of its high breathability and low chemically hazardous composition for garments with a large skin contact area, while its superior abrasion resistance

and unique coffee aroma will provide a more durable and unique experience compared to traditional synthetic leather footwear and bags. When applying it, designers should take into account the characteristics of sustainable coffee-ground synthetic leather and the environmental implications of sustainability and choose the appropriate process to maximize the value of the product.

## 4. Material Experiments

### 4.1. Experimental Procedure and Method

4.1.1. Specimen Preparation

The materials and methods are described in depth in Section 2 above. The different ways of preparing the base and the middle layer, as well as the different kinds of coating combinations, are explained. Innovative points are also presented. The next experimental samples are prepared and experimentally verified with reference to the sustainable coffee-ground synthetic leather preparation method and process in Section 2.2. Six examples of sustainable coffee-ground synthetic leather embodiments were prepared by setting different variables within the process steps. The variables were adjusted by setting different ratios of coating slurry components and the slurry density of the aqueous residue foam layer.

Example 1: An environmentally friendly water-based synthetic leather, comprising a top layer, an intermediate layer, and a base layer in order, from outside to inside. The said intermediate layer is a 1-layer adhesive layer. The said base layer is a base cloth, and the said surface layer is formed by the water-based residual material surface layer slurry coating. The said bonding layer is formed by the water-based residue bonding layer slurry coating. The said water-based residue top layer slurry configuration includes the following components: anionic aliphatic waterborne polyurethane emulsion (50%), silicone defoamer (0.5%), silicone wetting agent (1.5%), emulsifier (2.1%), coffee grounds (30%), isothiazolinone fungicide (1.6%), sodium o-phenylphenol mold inhibitor (1.2%), aziridine crosslinking agent (2%), and cellulose thickening agent (2.5%).

The said aqueous residue adhesive layer slurry configuration includes the following components: anionic aliphatic waterborne polyurethane emulsion (50%), coffee grounds (20%), isothiazolinone fungicide (1.6%), o-phenylphenol sodium mildew inhibitor (1.2%), aziridine crosslinking agent (2%), organic silicon wetting agent (1.5%), emulsifier (2.1%), and cellulose thickening agent (2.5%).

The production process of this embodiment includes the following steps: In step 1, grind the solid waste residue with a grinder until the solid waste residue has a particle size of 2 μm. In step 2, prepare an aqueous residue topping slurry. The above configuration components are dispersed at 500 rpm for 30 min according to the mass fraction and then added with cellulose thickener. The viscosity of the slurry is adjusted to 2500 cps and then dispersed at 4500 rpm for 20 min. In step 3, prepare an aqueous slag bonding layer slurry. The above configuration components are dispersed at 500 rpm for 30 min according to the mass fraction, and then a cellulose thickener is added. The viscosity of the slurry is adjusted to 35,000 cps, followed by a dispersion at 4500 rpm for 20 min. In step 4, the aqueous slag top layer obtained in step 2 is coated on the release paper and dried. Then, apply the water-based slag bonding layer pulp obtained in step 3; then laminate the base fabric and dry to obtain the environmentally friendly water-based synthetic leather.

Example 2: An environmentally friendly water-based synthetic leather, including a top layer, an intermediate layer, and a base layer from the outside to the inside in order. The said intermediate layer is a 2-layer foam layer. The said base layer is a base cloth, and the said surface layer is formed by a water-based residual material surface layer slurry coating. The said foaming layer is formed by the water-based residue foam-layer coating. Said water-based residue top layer slurry configuration includes the following components: anionic aliphatic waterborne polyurethane emulsion (50%), silicone defoamer (0.8%), coffee grounds (30%), isothiazolinone fungicide (1.6%), o-phenylphenol sodium mold inhibitor

(1.2%), aziridine crosslinking agent (2%), silicone wetting agent (1.5%), emulsifier (2.1%), and cellulose thickening agent (3.0%).

Said aqueous residue foaming layer slurry configuration includes the following components: anionic aliphatic waterborne polyurethane emulsion (50%), silicone modified foaming agent (5%), coffee grounds (30%), isothiazolinone fungicide (1.8%), o-phenylphenol sodium mold inhibitor (1.5%), aziridine crosslinking agent (2%), cellulose thickening agent (2.5%), and emulsifier (2.1%).

The production process of this embodiment includes the following steps: In step 1, grind the solid waste residue material with a grinder until the solid waste residue material has a particle size of 2 um. In step 2, prepare the aqueous residue topping slurry. The above configuration components are dispersed at 500 rpm for 30 min according to the mass fraction and then added with cellulose thickener. The viscosity of the slurry is adjusted to 3000 cps, followed by dispersion at 4500 rpm for 20 min. Step 3: Preparation of aqueous slag foam layer slurry. Add cellulose thickener after dispersing the above configuration components at 500 rpm for 30 min according to the mass fraction. The viscosity of the slurry was adjusted to 25,000 cps and then dispersed at 4500 rpm for 20 min. In step 4, the aqueous slag topping obtained in step 2 is coated on the release paper and dried. Pass the water-based residue foam layer slurry obtained in step 3 into the foaming machine, adjust the foam density to 0.8 kg/L, and coat the top layer and dry it. After drying, repeat the coating of water-based residue foam layer slurry, then laminate the base fabric and dry it to get environmentally friendly water-based synthetic leather.

Example 3: It differs from Example 1 in that the base layer is a water-based slag dry base. The said water-based crumb dry base consists of a base fabric impregnated by an impregnation tank with an impregnating solution, coated with a water-based crumb foam layer slurry with a foam density of 0.8 kg/L, dried, and wound. This example differs from the production process of Example 1 in that the following steps are added before step 4 of Example 1: after impregnating the base fabric with the impregnating tank containing the impregnating solution, the base fabric is coated with 0.8 kg/L of aqueous slag foam layer slurry, dried, and wound to form an aqueous slag dry base. In step 4, the laminated base fabric is changed to a laminated water-based slag dry base and dried to obtain an environmentally friendly water-based synthetic leather.

Example 4: It differs from Example 1 in that the base fabric is impregnated with an impregnating bath containing an impregnating solution and then extruded, ironed, and dried. The impregnating solution is a 1:1 configuration of 5% NaCl solution and nonfluorinated water repellent. This example differs from the production process of Example 1 in that the following steps are added before step 4 of Example 1: the base fabric is impregnated by the impregnation tank with impregnating solution and then extruded, ironed, and dried.

Example 5: It differs from Example 2 in that the base fabric is impregnated with an impregnating bath containing an impregnating solution and then extruded, blanched, and dried. The impregnating solution is a 1:1 configuration of 5% NaCl solution and nonfluorinated water repellent. This example differs from the production process of Example 2 in that the following steps are added before step 4 of Example 2: the base fabric is impregnated with an impregnating solution in the impregnation tank and then extruded, ironed, and dried.

Example 6: It differs from Example 1: the particle size of coffee grounds is 1 μm, and the rest remains the same.

### 4.1.2. Scanning Electron Micrograph of Samples

Prior to experimenting with the quality assessment parameters for the six preparations, a cross-sectional scan of the samples was carried out using the electron microscope device Phenom® at a temperature of 15–25 °C and a humidity of 40–60%, allowing us to identify the results of the different compositions of the materials in conjunction with the six preparations in 4.4.1, as shown in Figure 5.

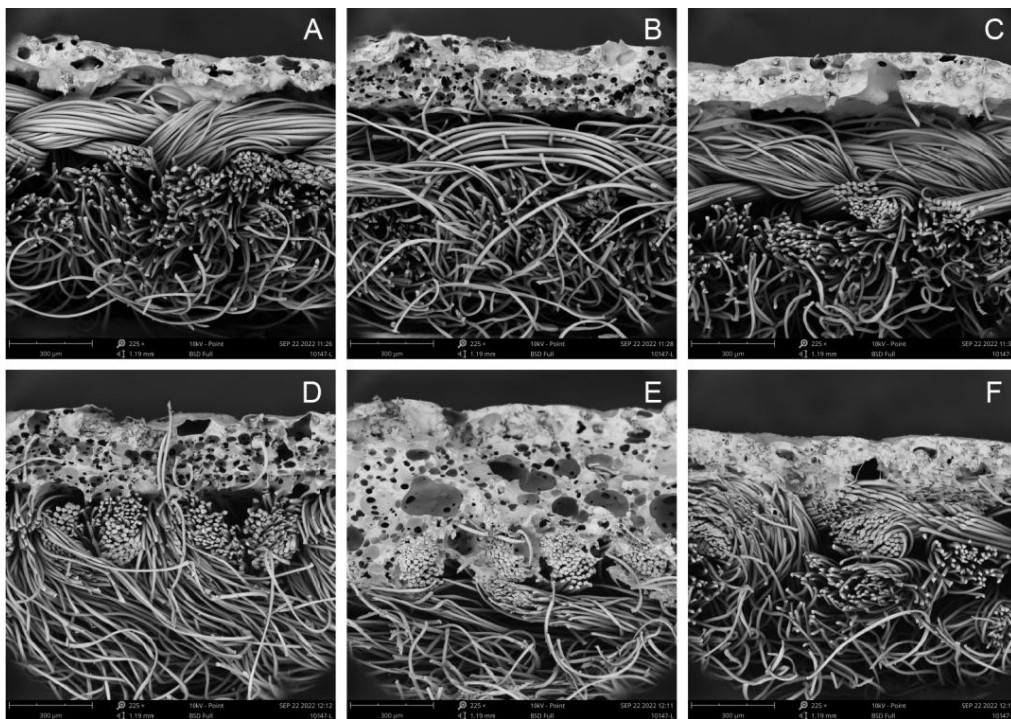

**Figure 5.** Cross-sectional scanning electron microscopy figures of the six samples, magnification of 225×. (**A**) Example 1. (**B**) Example 2. (**C**) Example 3. (**D**) Example 4. (**E**) Example 5. (**F**) Example 6.

### 4.1.3. Performance Testing

The environmentally friendly water-based synthetic leather of Example 1–6 and some synthetic leather samples were tested. The test contents are surface color fastness, Martindale rub test, peel load, constant temperature and flexing resistance, vapor permeability and moisture permeability, and softness. The above test content is the basic performance of synthetic leather for garments and soft bags.

### 4.2. Experimental Results

#### 4.2.1. Analysis of Wear Resistance Test Results

Abrasion resistance mainly assesses the solidity of the surface coating of the product, which is an important physical performance assessment index of synthetic leather. The performance test is carried out in accordance with the ISO 105-X12 rubbing color fastness standard and the GB/T19089 Martindale rub test standard.

As can be seen in Table 4, the rubbing color fastness of coffee on residue water-based synthetic leather is established under the influence of established pressure for dry and wet conditions. Sweat stain rubbing treatment is comparable to that of traditional solvent-based synthetic leather and water-based leather samples. The rubbing color fastness of Example 6 reached grade 5, surpassing the existing water-based leather samples. Applying the Martindale rub test, all the examples can achieve the performance of water-based leather samples. Among them, Examples 2, 3, 5, and 6 have better abrasion resistance, reaching 25,600 times without breakage.

**Table 4.** Color fastness to abrasion and Martindale rub test results.

| Samples | Friction Color Fastness (Grade) | | | Martindale Rub Test (Times) |
|---|---|---|---|---|
| | Dry Grinding | Wet Grinding | Sweat Stains | |
| Solvent-based garment leather samples | 4.5 | 4.5 | 4.5 | 12,800 Breakage |
| Sample of solvent-based upholstery leather | 4.5 | 4.5 | 4.5 | 25,600 Breakage |
| Water-based garment leather samples | 4.5 | 4.5 | 4.5 | 12,800 without breakage |
| Water-based upholstery leather samples | 4.5 | 4.5 | 4.5 | 25,600 Breakage |
| Example 1 | 4.5 | 4.5 | 4.5 | 25,600 Breakage |
| Example 2 | 4.5 | 4.5 | 4.5 | 25,600 No breakage |
| Example 3 | 4.5 | 4.5 | 4.5 | 25,600 No breakage |
| Example 4 | 4.5 | 4.5 | 4.5 | 25,600 Breakage |
| Example 5 | 4.5 | 4.5 | 4.5 | 25,600 No breakage |
| Example 6 | 5.0 | 5.0 | 5.0 | 25,600 No breakage |

### 4.2.2. Analysis of Vapor and Moisture Permeability Test Results

Vapor permeability and moisture permeability are two important performance indicators for measuring the comfort and hygiene of garment materials. Synthetic leather naturally suffers from poor vapor and moisture permeability. The results of the tests of the six examples are shown in Figure 6, referring to the ISO 20334:2011 vapor and moisture permeability standard.

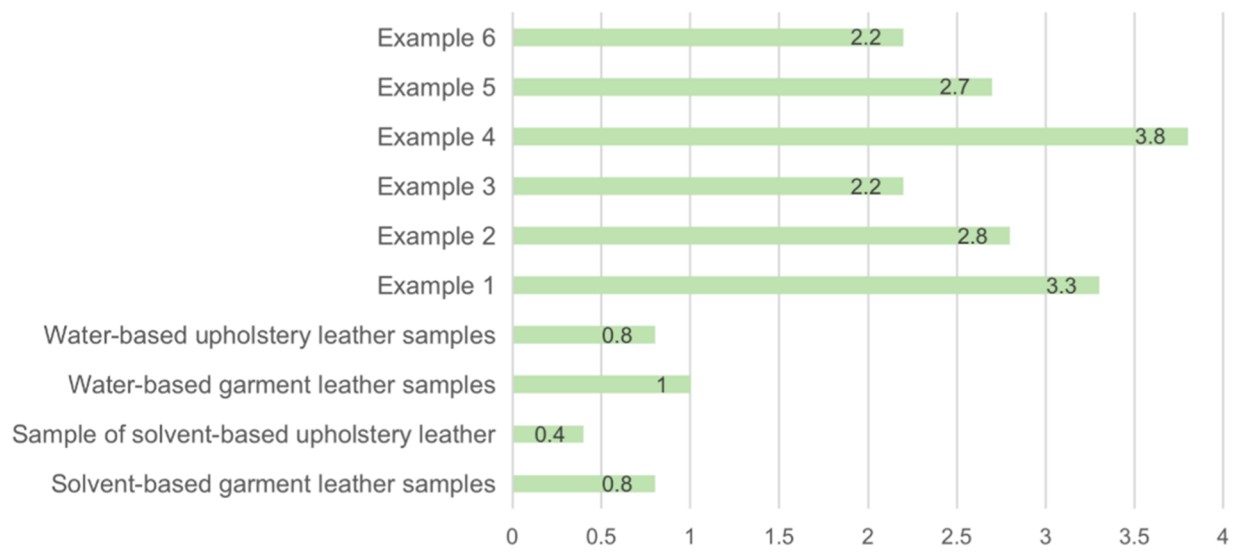

**Figure 6.** Vapor permeability and moisture permeability test results.

By applying the results of vapor permeability and moisture permeability of water-based synthetic leather samples with coffee grounds, they surpassed both traditional solvent-based synthetic leather and water-based leather samples. It can be seen that the addition of coffee grounds as solid waste, with its special pore structure characteristics of high specific surface area, has significantly increased the breathability of water-based synthetic leather.

### 4.2.3. Analysis of Softness Test Results

Please refer to the QB/T 5155 softness test method. The softness of the water-based synthetic leather samples with coffee grounds was explored and compared with traditional solvent-based synthetic leather and water-based leather, and the results are shown in

Figure 7. The overall softness of coffee-ground synthetic leather reached the softness level of traditional solvent-based and water-based soft-package leather.

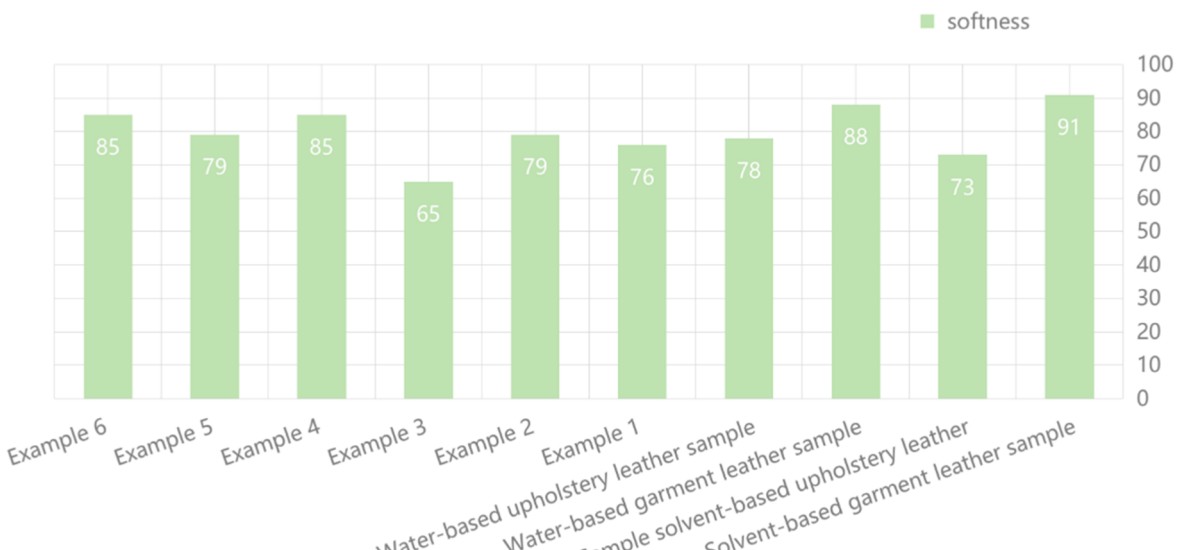

**Figure 7.** Softness performance test results.

4.2.4. Analysis of Peel Strength Test Results

The size of peel strength directly affects the quality and life of synthetic leather. The performance of coffee-ground synthetic leather of 1–6 examples was tested according to the 5.9 peel load standard in GB/T 8949. The results are shown in Figure 8. The peel strength performance of all six examples exceeded that of the water-based garment leather samples. Among them, the test results of Examples 1 and 6 performed better, with data exceeding 3.5 kg/3 cm.

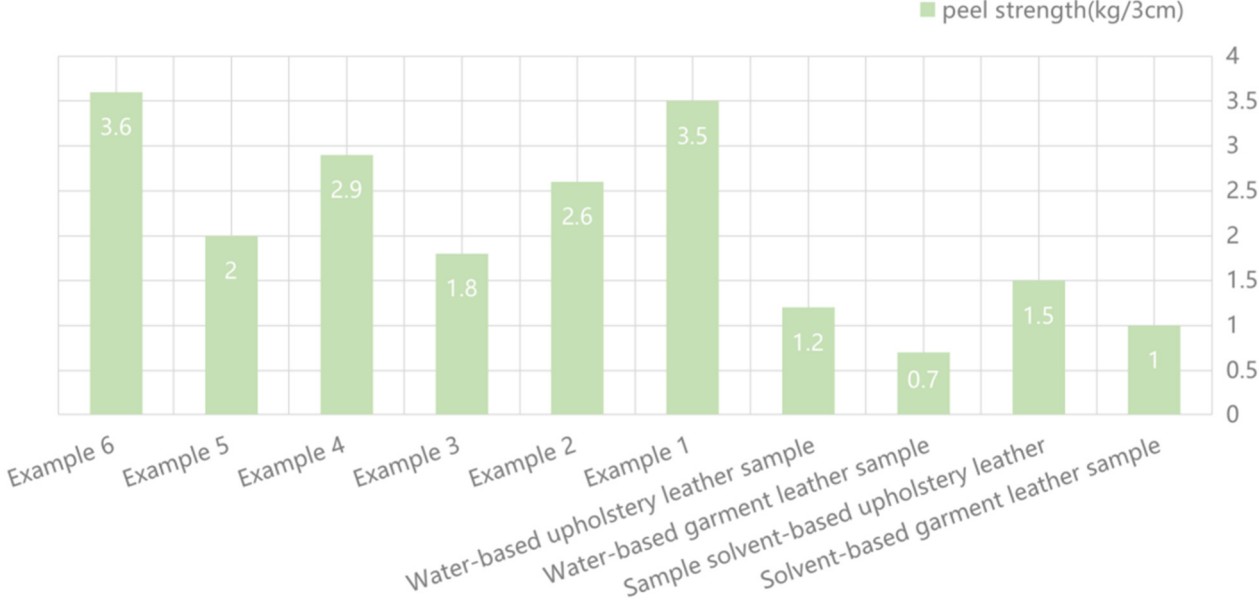

**Figure 8.** Peel strength performance test results.

The data of the above examples show that by using coffee grounds to replace fillers and other materials in the prior art, the products can fully achieve the performance of water-based synthetic leather for garments and bags, and even some of the examples and some performance indicators exceed the existing water-based synthetic leather.

## 5. Conclusions

In recent years, the development and application of coffee grounds have become a new hot spot for research and exploration in the field of sustainability. A study of the literature reveals that these developments are currently focused on fuels, adsorbents, blended yarns, color printing, dyeing, etc., and there is still a gap in the preparation methods and application research of synthetic leather. This paper summarizes the preparation method of water-based synthetic leather by reusing solid waste coffee grounds as fillers and developing a new environmentally friendly production process of water-based synthetic leather, testing its properties, such as color fastness to dry and wet rubbing, and designing applications of sustainable coffee grounds. Synthetic leather in garments and apparel products is analyzed. The experimental results show that the material fully achieves the performance of waterborne garments and luggage synthetic leather and can provide breathability close to that of genuine leather. In addition, it can completely solve the problem of chemical pollution caused by DMF in solvent-based synthetic leather in the current market and eliminate the DMF recycling process. This makes the whole production process energy-saving and environmentally friendly, which is an innovative and sustainable product that can be applied to the apparel industry in the future. Of course, this method is not limited to the application of coffee grounds but can also be extended to the recycling and reuse of more solid waste impurities, and as research continues, it will be more widely developed and applied in the field of textiles, turning environmental problems into opportunities.

**Author Contributions:** Conceptualization, Y.T.; Formal analysis, J.W. and X.H.; Project administration, Y.T.; Resources, S.Z.; Supervision, X.L.; Writing–original draft, J.W. All authors have read and agreed to the published version of the manuscript.

**Funding:** This research received no external funding.

**Data Availability Statement:** No applicable.

**Conflicts of Interest:** The authors declare no conflict of interest.

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
