# Peer review of "Research on the Preparation and Application of Synthetic Leather from Coffee Grounds for Sustainable Development"

_sustainability, doi:10.3390/su142113971_

Round 1

Reviewer 1 Report

Review of sustainability-1903202

This is a manuscript about an innovative product development, multidisciplinary intersection of fashion design with material science and environmental management in order to achieve sustainability. There are several issues to be revised and added, as follows:

  1. Please add cross-sectional scanning electron microscopy figures of the samples prepared in this manuscript (Section 4.1.1).
  2. Line 41: Please add these papers, since coffee grounds are also utilized for synthesis of advanced materials for environmental sustainability, such as photocatalyst and photothermal materials.
  • Food and Bioproducts Processing 121 (2020) 193-201 https://doi.org/10.1016/j.fbp.2020.02.008
  • Journal of the Taiwan Institute of Chemical Engineers  (2022) 104259 https://doi.org/10.1016/j.jtice.2022.104259

  1. Line 26: [3-4]
  2. Line 39: …greenhouse gases… --> delete the quote and end quote symbols.
  3. Line 41: [6-7]
  4. Line 61-64: Add information about the name of the manufacturers or suppliers, and also their countries.
  5. Table 1: Please add new column about the mass of the “copies”.
  6. Line 67-69: Middle layer is comprised of: (1) one adhesive layer, or (2) two foaming layers, or (3) one foaming layer with one adhesive layer.
  7. Line 67-69: Please illustrate also about what do you mean in line 67-69, so that the readers can easily understand this manuscript.
  8. Line 76: …waterborne… --> lowercase w.
  9. Table 2: Please add new column about the mass of the “copies”.
  10. Line 93: 0.1-5.0 µm --> use µ sign (micro), not the letter u.
  11. Table 3: …waterborne… --> lowercase w.
  12. Table 3 again: …waterborne… --> lowercase w.
  13. Table 3 again:Cross-linking… --> uppercase C.
  14. Line 105: 20-30 °C
  15. Line 106: 0.1-5.0 µm --> use µ sign (micro), not the letter u.
  16. Line 107: …waterborne… --> lowercase w.
  17. Line 110: …400-600 rpm…
  18. Line 112: Please separate “5000” and “cps” with a space.
  19. Line 112: …4000-5000 rpm…
  20. Line 112: Please separate “30” and “min” with a space
  21. Line 116: …waterborne… --> lowercase w.
  22. Line 118: …400-600 rpm…
  23. Line 120: …4000-5000 rpm…
  24. Line 120: Please separate “30” and “min” with a space
  25. Line 122: …waterborne… --> lowercase w.
  26. Line 125: …400-600 rpm…
  27. Line 125: Please separate “30” and “min” with a space
  28. Line 126: Please separate “5000” and “cps” with a space.
  29. Line 128: …4000-5000 rpm…
  30. Line 146: …wet base. It is then dried to…
  31. Line 168: …waterborne… --> lowercase w.
  32. Line 260: [16-17]
  33. Line 279-304: Please split this long paragraph to be 2-3 shorter paragraphs.
  34. Line 305-332: Please split this long paragraph to be 2-3 shorter paragraphs.
  35. Line 279-304, 305-332, 333-358: Please change “r/min” to be “rpm”.
  36. Line 279-304, 305-332, 333-358: Please change “Waterborne” to be “waterborne” --> lowercase w
  37. Line 279-304, 305-332, 333-358: Please separate number and “min” with a space.

  1. Line 279-304, 305-332, 333-358: Please separate number and “kg/L” with a space.
  2. Line 279-304, 305-332, 333-358: Please write micrometer as “µm”, not “um”.
  3. Line 345: NaCl --> uppercase N and C
  4. Line 357: 1 µm
  5. Line 377: By applying… --> start this sentence with “By”
  6. Line 408: Please separate “3.5” and “kg” with a space.
  7. Line 408: What do you mean with 3.5 kg/3c? Please justify.

Reviewer 2 Report

Tian and his coauthors prepared and characterized synthetic leather filled with coffee residues. The authors also compared their materials with some conventional ones.

The paper is relatively well written and has an enough scientific relevance. I suggest that the article could be published after profound improvements. The small number of quality parameters evaluated seems to me to be a significant issue.

However, the organization of the items and the excess of irrelevant information are the most relevant problems in my opinion. I suggest that item 3 be deeply summarized and incorporated into the introduction (item 1). I also suggest that item 2 be summarized and incorporated into item 4. The work carried out is not a literature review. This work is an original article due to the experimental work performed. Please follow the most usual format for articles in this area, where there is an introductory item dealing with the literature review and the importance of the study. After this, a M&M item, which describes the used materials and the techniques performed. Also, the prior preparation of some raw materials may be relevant in some cases (such as the leather in the present study).

I didn't understand the unit used in tables 1 and 2 (i.e. copies). It would be possible to inform the composition in either mass or volume.

Discussion of the results practically does not exist. There are no consistent comparisons with similar works recently published. There are no comments on elucidations brought by other studies, which could support the conclusions of the present study.

Reviewer 3 Report

In this research, preparation and application of synthetic  leather from coffe grounds is reported. The general idea is interesting and may be attractive to a broad audience. The manuscriptis in the scope of sustainability. However, there exist many points to  be addressed.

1. Sections 2.1 and 2.1.1 have the the same name; I think, it is necessary to adequate.

2. The material composition of each layer is displayed in tables, then it is not necessary to mention widely in the  text. In tables use appropriate units.

3. When mentioning quantities, the units should be separated from the numbers.

4. All figures should be mentioned in the text.

5. Section 2.2.4 is confuse, mainly the second paragraph; section 4.1.1 may be enough.

6. Results are only presented, a discussion is needed.

Round 2

Reviewer 1 Report

Review of sustainability-1903202-v2

The authors have put a nice effort to improve this manuscript significantly. The manuscript can be accepted now.

Note: Please use only American English, or only British English, but not both. Because American English is used mainly in this manuscript, there are some British English words to be revised, as follows:

Line 14: standardized

Line 58: analyzing

Line 293: maximize

They can be revised during the proofreading stage, thanks.

Reviewer 2 Report

honestly, I don't see a relationship with literature. The work continues without discussion of results, which seems to me to be something very strange. On the other hand, the work is well written and the organization of items has improved in this latest version. In my opinion, the work can be accepted, if the editor sees interest in this subject.

Reviewer 3 Report

Authors have addressed all the suggestions carefully. The new version of the paper has been improved and is suitable for publication.